# Using Mean Arterial Pressure in Hypertension Diagnosis versus Using Either Systolic or Diastolic Blood Pressure Measurements

**DOI:** 10.3390/biomedicines11030849

**Published:** 2023-03-10

**Authors:** Heba Kandil, Ahmed Soliman, Norah Saleh Alghamdi, J. Richard Jennings, Ayman El-Baz

**Affiliations:** 1Bioengineering Department, University of Louisville, Louisville, KY 40292, USA; 2Information Technology Department, Faculty of Computers and Informatics, Mansoura University, Mansoura 35516, Egypt; 3Department of Computer Sciences, College of Computer and Information Sciences, Princess Nourah Bint Abdulrahman University, P.O. Box 84428, Riyadh 11671, Saudi Arabia; 4Departments of Psychiatry and Psychology, University of Pittsburgh, Pittsburgh, PA 15260, USA

**Keywords:** hypertension, tortuosity, cerebral, blood vessels, MAP, neural networks, systolic, diastolic, SVM, KNN, linear discriminant, logistic regression

## Abstract

Hypertension is a severe and highly prevalent disease. It is considered a leading contributor to mortality worldwide. Diagnosis guidelines for hypertension use systolic and diastolic blood pressure (BP) together. Mean arterial pressure (MAP), which refers to the average of the arterial blood pressure through a single cardiac cycle, can be an alternative index that may capture the overall exposure of the person to a heightened pressure. A clinical hypothesis, however, suggests that in patients over 50 years old in age, systolic BP may be more predictive of adverse events, while in patients under 50 years old, diastolic BP may be slightly more predictive. In this study, we investigated the correlation between cerebrovascular changes, (impacted by hypertension), and MAP, systolic BP, and diastolic BP separately. Several experiments were conducted using real and synthetic magnetic resonance angiography (MRA) data, along with corresponding BP measurements. Each experiment employs the following methodology: First, MRA data were processed to remove noise, bias, or inhomogeneity. Second, the cerebrovasculature was delineated for MRA subjects using a 3D adaptive region growing connected components algorithm. Third, vascular features (changes in blood vessel’s diameters and tortuosity) that describe cerebrovascular alterations that occur prior to and during the development of hypertension were extracted. Finally, feature vectors were constructed, and data were classified using different classifiers, such as SVM, KNN, linear discriminant, and logistic regression, into either normotensives or hypertensives according to the cerebral vascular alterations and the BP measurements. The initial results showed that MAP would be more beneficial and accurate in identifying the cerebrovascular impact of hypertension (accuracy up to 95.2%) than just using either systolic BP (accuracy up to 89.3%) or diastolic BP (accuracy up to 88.9%). This result emphasizes the pathophysiological significance of MAP and supports prior views that this simple measure may be a superior index for the definition of hypertension and research on hypertension.

## 1. Introduction

Hypertension is a disease that contributes to the mortality rate of millions of people in the world [1]. The 2017 guidelines [2] define diagnosis of hypertension using measurements of both components: systolic blood pressure (BP) and diastolic BP. For instance, a patient is considered to be in Hypertension Stage II if the individual has a systolic reading of BP ≥ 140 mmHg or a diastolic reading of BP ≥ 90.

Whereas both systolic and diastolic measurements remain essential for the diagnosis and the treatment, clinical research suggests that in patients above 50 years old in age, systolic BP may be a more important predictor of adverse events, whereas in patients under 50 years old, diastolic BP may be slightly more predictive [3]. High systolic BP in the absence of high diastolic BP is quite prevalent in the elderly, while other clinical studies suggest the essential role of diastolic BP in younger people [4,5,6,7,8,9,10].

Perhaps, in part to the variable importance assigned to systolic and diastolic BP in the literature, Kundu et al. [11] suggested that mean arterial pressure (MAP) should be used to better predict the blood pressure compared to the usage of systolic BP or diastolic BP separately. MAP, which refers to the average of the arterial blood pressure through a single cardiac cycle, may thus be a better alternative for the diagnosis and statistical analysis of blood pressure. The value of using MAP was shown both when examining the association of blood pressure with potential causative factors [11], as well as in an enhanced ability to detect mild cases of hypertension.

One important seeming consequence of hypertension is the alterations in the cerebrovascular that include the loss of vessels as well as changes in their morphology [12]. Such changes may relate to the now well-known impact of hypertension on cognitive function and risk of dementia [13]. It is interesting to refer to the fact that these cerebral changes may start to develop before the systemic onset of hypertension. Tracking the progress of these alterations would help clinicians to take proactive and preventative steps to avoid the progression of the disease and its related complications [14].

In the current research, we focus on measures of the cerebrovasculature and ask whether MAP shows a stronger association with these measures than either systolic BP or diastolic BP taken separately. Our study presented extensive experimentation that emphasized the pathophysiological significance of MAP and supported prior views that recommended using MAP as a superior index for the definition of hypertension and related research.

## 2. Materials and Methods

Methodology started with classifying data into three different datasets. One dataset was classified into normotensive and hypertensive subjects based on systolic BP. The second dataset was classified into normotensive and hypertensive subjects based on diastolic BP. Finally, the third dataset was classified based on MAP values of the subjects. The following step was studying the correlation between each dataset and the vascular changes that predict and diagnose hypertension to see which dataset (and which measurements: systolic BP, diastolic BP, or MAP) would give higher prediction accuracy. To find the vascular changes, the following procedure was used. First, magnetic resonance angiography (MRA) data were prepared and preprocessed to eliminate noise and bias effects and to improve the quality of data. Second, clean data were segmented to extract the cerebrovasculature from each subject using a 3D adaptive region growing connected components algorithm. After that, cerebrovascular features which describe the changes in cerebral vasculature were extracted. These features are able to predict the potentiality of hypertension development. Feature vectors were then constructed to be fed into different classifiers. Finally, data were classified into either hypertensive or normotensive according to alterations of the cerebrovascular features. More details of each step are presented in the following subsections. Figure 1 shows the different steps of the proposed framework presented in this study.

### 2.1. Data Preparation

A data set of 342 subjects of MRA was used in the experimentation. Data were acquired by the University of Pittsburgh and approved by IRB according to the relevant guidelines and regulations [15]. Participants of the study were chosen to be middle-aged adults (age 35–62 years, mean= 51 ± 6.6) that do not suffer from any of the following conditions: cancer, pregnancy, ischemic coronary artery disease, chronic kidney or liver diseases, diabetes mellitus, multiple sclerosis, strokes, epilepsy, serious head injury, brain tumor or mental illness. Additionally, participants did not use any prescribed medications for hypertension. Participants provided written consent before getting involved in the study.

Blood pressure readings were taken for participants 4 times during 3 visits within 2 weeks. Among other procedures, visits 1 and 2 included BP readings. In visit 3, participants were asked to sit down and relax while supporting their back and arms for at least 5 min before trained assistants took their BP readings twice, separated by 1 min. The auscultatory technique was used to measure BP of participants with cuff size fitted according to arms circumferences. All four readings were averaged and used as the final resting BP measurement for each individual. This study followed the 2017 guidelines [2] in classifying data. The guidelines defines a normal individual to have a systolic BP < 120 mmHg and diastolic readings < 80 mmHg. Individuals are defined to suffer from hypertension stage 1 if they have a systolic BP of 130–139 mmHg or diastolic readings of 80–89 mmHg. For individuals with hypertension stage 2, the readings should be ≥140 mmHg or diastolic readings ≥90 mmHg. Based on that, a systolic hypertensive subject is defined to have a systolic reading >130 mmHg, whereas a diastolic hypertensive patient should have a diastolic reading >80 mmHg.

The dataset of 342 subjects was partitioned as follows: 116 subjects (see Table 1 for demographics information of this group) were selected according to systolic BP measurements (68 were normotensive, and 48 were hypertensive), and 226 subjects (see Table 2 for demographics information of this group) were selected according to diastolic BP measurements (143 were hypertensive, and 83 were normotensive).

To solve the challenge of the non-balanced dataset and increase the accuracy of the results, new synthetic normal samples and hypertensive samples were generated to obtain balanced data sets. The details of the procedure used to generate synthetic data samples are presented in Section 2.4.

To categories data based on MAP, we used the equation that follows: ( MAP = (1/3) * systolic BP+ (2/3) * diastolic BP) to calculate MAP for each subject. Then, based on the categorization of MAP suggested in [11], MAP values were classified as optimal, normal, high normal, grade 1 hypertension (mild), grade 2 hypertension (moderate), or grade 3 hypertension (severe). An optimal MAP value is <93.33, a normal MAP value is in the range of (93.33–99.00), a high normal value should be in the range of (99.01–105.67), grade 1 hypertension range is (105.68–119.00), grade 2 hypertension range is (119.01–132.33), and grade 3 hypertension MAP values are ≥132.34. In this study, we considered normal MAP values as one class, and high normal, hypertension grade 1, hypertension grade 2, hypertension grade 3 as the other class. Thus, 161 out of 342 subjects (see Table 3 for the demographics information of this group) were used in MAP-based experiments. We did not include the optimal category of MAP values in our experimentation because there is no corresponding category in the experiments that considers either systolic BP or diastolic BP.

### 2.2. Cerebrovascular Segmentation

A preprocessing step was used to clean MRA data by removing noise, correcting bias, and increasing homogeneity among each volume’s slices using a bias correction algorithm [16], and a 3D generalized Gauss–Markov random field (GGMRF) model [17]. A skull-stripping approach was then used to get rid of the brain’s fat tissues and only retain brain tissues. The approach used the Markov–Gibbs random field model combined with a geometric deformable model (brain isosurface) to maintain cerebrovascular topology while delineating the cerebrovasculature.

The segmentation of cerebral vasculature started with a linear combination of discrete Gaussians to estimate the marginal probability density of MRA voxel values for vessels and other brain tissues [18,19]. This results in an initial delineated vasculature that misses some details of the cerebrovasculature such as tiny blood vessels. To enhance the initial segmented vasculature, a 3D adaptive segmentation method was used [20]. This algorithm works by dividing each MRA slice into a set of connected components. A search window of adaptive size was centered around each component in the set and a new separation threshold was calculated as $T=\frac{\mu_{b}+\mu_{o}}{2}$, where $\mu_{b}$ is the intensity average of cerebral vessels and $\mu_{o}$ is the intensity average of other cerebral tissues. Additionally, a seed-generation refinement procedure was applied to detect potential seeds within regions with a high potential to contain small cerebral vessels that might have been missed in the initial delineated vasculature. Finally, a 3D region growing connected components algorithm was used to obtain the final vasculature. This algorithm achieved 92.23% Dice similarity coefficient, 94.82% sensitivity, and 99.0% specificity. A preprocessing sample is shown in Figure 2, and a segmentation sample output is shown in Figure 3.

### 2.3. Cerebrovascular Features Extraction

The delineated cerebral vascular tree was used to track and estimate the cerebral features to be used in the classification process. The features that were chosen and estimated are the change of diameters of blood vessels and the alterations of the vessels’ tortuosity. Clinical research supported the correlation between hypertension and alterations of the cerebral vessels diameter size and tortuosity [21,22,23].

To quantify the vascular features, the approach presented in [12] was followed. To estimate the changes in diameter size of the blood vessels, medians of vascular radii were estimated for each MRA volume by generating a distance map for the delineated cerebral tree. Then, the cumulative distribution function (CDF) of the vascular radii was obtained as the cumulative distribution of probability density function (PDF). CDF provides a probability estimate for blood vessels that exist at or below a specific vascular diameter point. A CDF value defines the average of vascular diameters in an MRA volume [12].

Similarly, to estimate the changes in blood vessels tortuosity, estimations of Gaussian and mean curvatures were calculated. The mean curvature was estimated by the following equation: (k1 + k2)/2, where k1 and k2 are the principal curvatures. The Gaussian curvature was estimated by the equation: k1 × k2 [12]. A feature vector was built to describe the cerebral vascular changes of every subject. Each feature vector includes 17 values, of which 13 values were used to describe the vascular diameters change ( 11 values are PDF bins that correspond to every diameter radius, and the remaining two values are for median and average of vascular radii), and the rest of the feature vector values were used to describe the change in tortuosity of the blood vessels (estimated by averages and medians and of Gaussian and mean curvatures). A sample of the feature vectors of four different subjects is shown in Table 4.

### 2.4. Synthetic Data Generation

The current study uses the synthetic minority over-sampling technique (SMOTE) [24] to generate synthetic samples to compensate for the dataset imbalance between the number of normotensive and hypertensive subjects. SMOTE is a very effective and simple technique that is used in many studies [25,26,27] to generate synthetic data samples with similar features to the original data sets. SMOTE works in feature space, not in data space. In order to obtain a new sample in the minority class, a random point is selected from the class along with the *k*-nearest neighbors of that point. Every feature in the feature space is then multiplied by a new normalized random weight in the range [0–1]. The procedure repeats with the new random weights for all features of the selected *k*-nearest neighbors. A new synthetic sample is then estimated as the accumulation of multiplications between generated random weights and feature vectors. This technique guarantees that the training process is balanced enough to build an unbiased classification model. That model uses the synthetically generated data that have the common characteristics of the original data without being a linear combination of any subset of the samples in the minority class.

### 2.5. Design of Experiments

Different measurements were applied to categorize data into hypertensive and normotensive classes. In this research, we had two stages of experimentation (Figure 4). In the first stage, three separate experiments were implemented (Table 5 and Table 6). Each separate experiment includes two sub-experiments. The first sub-experiment considered the systolic BP only to select hypertension class members, whereas the second sub-experiment depended on diastolic BP measurements only to select hypertensive subjects.

The first experiment was performed using the original dataset, which was imbalanced. The systolic sub-experiment was performed using a data set of 116 subjects, of which 68 were normotensive and 48 were hypertensive, whereas the diastolic sub-experiment was conducted using a total of 226 subjects, of which 83 were normotensive and 143 were hypertensive.

The other two experiments were performed using both the original data and the new synthetic data samples. The synthetic samples were generated to enlarge the original data set and to ensure it was balanced. In the second experiment (systolic sub-Experiment), 20 synthetic hypertensive samples were generated, making a total of 136 subjects in the balanced dataset, of which 68 were hypertensive and 68 were normotensive. Similarly, in the diastolic sub-experiment, 60 normotensive samples were generated, making a total of 286 subjects, of which 143 were hypertensive and 143 were normotensive. For the third experiment, more synthetic data samples were generated to increase the number of subjects tested in both sub-experiments to 600 subjects (300 were hypertensive, 300 were normotensive).

In the second stage, we also conducted three experiments similar to the experiments in stage 1, but this time based on MAP categorization (Table 7). The first experiment included 161 subjects (107 normotensive, 54 hypertensive). The second experiment used a balanced data set of 108 subjects (54 normotensive, 54 hypertensive). The third experiment was conducted after enlarging and balancing the original data set by the incorporation of more synthetic samples. Two trials were conducted in the third experiment, one using 214 subjects (107 normotensive, 107 hypertensive), and the other trial using 300 subjects (150 normotensive, 150 hypertensive).

Data are classified using various classifiers (linear and non-linear) with various parameters and scenarios of validation. Classification accuracy is computed as follows:(1)$$Accuracy=(\frac{number\;\;of\;\;true\;\;classified\;\;samples}{number\;\;of\;\;total\;\;test\;\;data})\times 100$$

### 2.6. Classifiers

Several classifiers were used in this study, such as SVM, decision trees, linear discriminant, logistic regression, and KNN. Support vector machine (SVM) [28] can be used when the data have exactly two classes. An SVM classifies data by finding the best hyperplane that separates all data points of one class from those of the other class. The best hyperplane for an SVM means the one with the largest margin between the two classes. Margin means the maximal width of the slab parallel to the hyperplane that has no interior data points. The support vectors are the data points that are closest to the separating hyperplane; these points are on the boundary of the slab. Decision trees, or classification trees and regression trees, predict responses to data. To predict a response, follow the decisions in the tree from the root (beginning) node down to a leaf node. The leaf node contains the response. Classification trees give responses that are nominal, such as ‘true’ or ‘false’. Regression trees give numeric responses. Discriminant analysis is a classification method. It assumes that different classes generate data based on different Gaussian distributions. To train (create) a classifier, the fitting function estimates the parameters of a Gaussian distribution for each class. To predict the classes of new data, the trained classifier finds the class with the smallest misclassification cost. Linear discriminant analysis is also known as the Fisher discriminant, named for its inventor, Sir R. A. Fisher. Logistic regression is a classification technique used in machine learning. It uses a logistic function to model the dependent variable. The dependent variable is dichotomous in nature, i.e., there could only be two possible classes (e.g., either the person is hypertensive or not). The k-nearest neighbors algorithm, also known as KNN or k-NN, is a non-parametric, supervised learning classifier, which uses proximity to make classifications or predictions about the grouping of an individual data point. While it can be used for either regression or classification problems, it is typically used as a classification algorithm, working off the assumption that similar points can be found near one another.

## 3. Results

The MATLAB R2017a built-in classification learner was used in the study experimentation to allow for using different classifiers with various parameters and validation scenarios. The partitioning of data between training, testing, and validation is done randomly between different patients. In the first stage experiments, the accuracy of classification recorded in the first experiment was unpromising, neither in the systolic sub-experiment nor in the diastolic sub-experiment. The best achieved results for the various classifiers recorded an accuracy of 60.0%. It is believed this may have been due to the fact that the original data set was not large enough and was not balanced.

On the other hand, the classification accuracy achieved in the second experiment was more promising as given in Table 8 and Table 9 for systolic sub-experiment and diastolic sub-experiment, respectively. Systolic sub-experiment recorded a classification accuracy of 85.0% by a feed-forward neural network (NN) with two hidden layers of sizes: 10, 8, and a leave-one-subject-out (leave-one-out) validation setup. The same accuracy was recorded using the random undersampling boosting ensemble (RUBoosted) classifier as well, using a 15.0% hold-out validation setup.

The diastolic sub-experiment, however, recorded an accuracy of 72.4% by the decision tree classifier with a 20-fold validation setup. The third experiment achieved the highest classification accuracy in both systolic and diastolic sub-experiments as given in Table 10 and Table 11, respectively. The best recorded accuracy in the systolic sub-experiment was 89.3% by the logistic regression classifier and a 25.0% hold-out validation setup as given in Table 10, whereas the best recorded accuracy for the diastolic sub-experiment was 88.9% by a feed-forward NN with two hidden layers of sizes 13,9, and a leave-one-out validation setup (Table 11).

For the second stage experiments, the best achieved accuracy in the first experiment (161 subjects) was 87.5% using a K-nearest neighbor (KNN) classifier with a Euclidean distance metric and a 10.0% hold-out validation setup (Table 12). This was very interesting because in this experiment, the dataset was not balanced; however, the classification accuracy was promising. The best achieved accuracy in the second experiment (108 subjects (balanced)) was 71.4% using either a KNN with a cosine metric or a support vector machine (SVM) with Gaussian kernel and 20.0% hold out validation scenarios, Table 13. In the third experiment, the best recorded accuracy was 95.2% using AdaBoost ensemble with a 10.0% hold-out validation scenario, Table 14 and Table 15.

In this manuscript, we included accuracy values that are ≥70.0% only. However, we included the results of classification using the SVM classifier in all experiments (except for Table 12 because it was less than 70.0%) for the purpose of comparison between different experiments as given in Table 16. As shown in the table, MAP-based classification recorded the best accuracy of 90.0%. The systolic-based experiment recorded the second best accuracy of 84.3%, whereas the diastolic-based experiment recorded 76.7%. Results support the significance of using MAP in hypertension prediction and prove its correlation to the cerebral vascular changes that occur prior to and during the development of hypertension.

## 4. Discussion

According to the 2017 guidelines for hypertension [2], deciding whether an individual is normal or having hypertension is based on blood pressure readings of either systolic BP, diastolic BP, or both. Systolic blood pressure defines how much pressure the blood is exerting against the artery walls when the heart pumps the blood out. Diastolic blood pressure defines how much pressure the blood is exerting against the artery walls when the heart is resting between beats. A normal individual should have both systolic and diastolic BP reading under specific values (<120/<80 mmHg). However, an individual is diagnosed to have hypertension stage 2 if the systolic blood pressure is ≥140 mmHg or the diastolic BP is ≥90 mmHg. So, only one reading (either systolic or diastolic) is enough in the severe hypertension diagnosis. Many hypertensive patients have both their systolic and diastolic readings exceed the normal ranges. However, in many other cases, the individual would have one of the measurement components (either systolic or diastolic) in the normal range, while the other component exceeds the normal range. In this case, the overall diagnosis is the existence of the disease. For instance, it is quite prevalent in the elderly to have high systolic BP while their diastolic blood pressure is stable and normal [4,5,6,7,8,9,10]. In contrast, in younger individuals, the diastolic blood pressure plays an important role in affecting their health. Some studies investigated the importance of one component over the other in affecting people’s health based on their ages [3,4,5,6,7,8,9,10].

In this study, we investigated the importance of the systolic component and diastolic component separately in hypertension diagnosis. The aim was to find out whether one component would be of more value in hypertension detection than the other. Additionally, we studied the usefulness of MAP in hypertension diagnosis as a measurement that benefits from both components (the systolic and the diastolic) rather than taking only one component into consideration as explained in the previous few lines. The data set used in the study experimentation consisted of individuals with a wide range of ages (35–62 years) to test whether the age would have a role in the importance of component over the other in hypertension diagnosis. Preliminary results showed that using the systolic BP component for people aged 35–62 years can be more predictive than using diastolic BP (see Table 8, Table 9, Table 10 and Table 11). Additionally, the results proved that using MAP would enhance the accuracy of hypertension diagnosis more than using single components (see Table 16).

Experiments conducted in the study were designed to perform supervised classification tasks. Labeled data were introduced to various classifiers to measure the accuracy of differentiating between normal and hypertensive individuals based on both blood pressure measurement and cerebral vascular changes. The cerebral vascular changes data were incorporated in the classification tasks to study the correlation between hypertension-related cerebrovascular changes and systolic BP, diastolic BP, and MAP. The results showed that these cerebral changes affect systolic and diastolic components to quite different degrees. Generally, as seen in Table 8, Table 9, Table 10 and Table 11), the classification accuracy of the systolic-based experiment was better than the diastolic-based experiment. This could be evidence that the cerebral vascular changes associated with hypertension development might affect the systolic BP component with a higher degree than the diastolic BP component. However, using MAP which incorporates information from both the systolic and diastolic BP components would provide more accurate correlations between cerebrovascular changes and hypertension detection.

To the best of our knowledge, this is the first study to analyze the role of systolic BP, diastolic BP, and MAP independently and their correlations to changes in the cerebral vasculature. One limitation in our study is that it is very hard and expensive to obtain MRA scans for individuals over long periods of time so that we can track the cerebral changes easier. That was the main reason that the dataset is not big enough and that we needed synthetic data samples to enlarge the dataset. However, we will work on recruiting more individuals in the future to solve this issue. Additionally, MRA scans are expensive. Thus, we recommend to currently limit their usage in hypertension prediction for people with high potential of developing hypertension due to family history, for example. Another limitation is that the conclusion of this study cannot be followed if the underlying study is targeting studying people who suffer from systolic-only hypertension or diastolic-only hypertension because in such cases, we need to study only one component (either systolic BP or diastolic BP) and not both. So, MAP, which incorporates information from both components, is not a good measure in this case.

## 5. Conclusions

This study investigated the significance of systolic BP versus diastolic BP in defining cerebrovascular sequelae of hypertension and compared that to the significance of using MAP. Preliminary results indicated that systolic BP might be more predictive of hypertensive association with cerebrovascular indices than diastolic blood pressure. However, using MAP values that incorporate information from both systolic and diastolic BP recorded the highest predictability in detecting hypertension-related vascular alteration than using systolic BP or diastolic BP separately. This result emphasizes the pathophysiological significance of MAP and supports prior views that this simple measure may be a superior index for the definition of hypertension and research on hypertension. Future plans include collecting more data over longer periods of time to allow for tracking the changes of the cerebral vascular and their impact on developing hypertension and to test the proposed methodology with more original data to enhance the accuracy and reliability of the results.

## Figures and Tables

**Figure 1 biomedicines-11-00849-f001:**
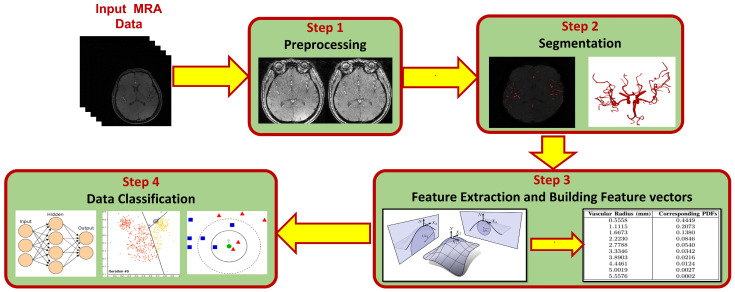
A block diagram showing modules of the proposed framework for classifying hypertension data.

**Figure 2 biomedicines-11-00849-f002:**
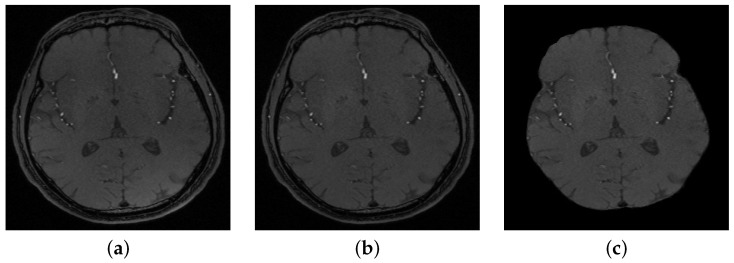
(**a**) A sample of a 2D original slice, (**b**) after bias correction, and (**c**) after skull stripping.

**Figure 3 biomedicines-11-00849-f003:**
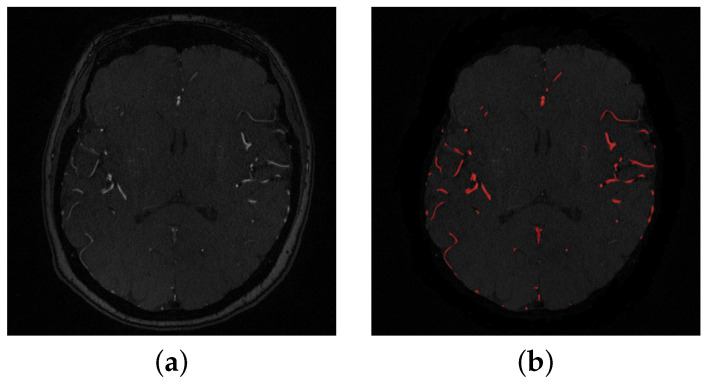
(**a**) A sample of a 2-D original slice, (**b**) Segmented slice where detected blood vessels are colored in red.

**Figure 4 biomedicines-11-00849-f004:**
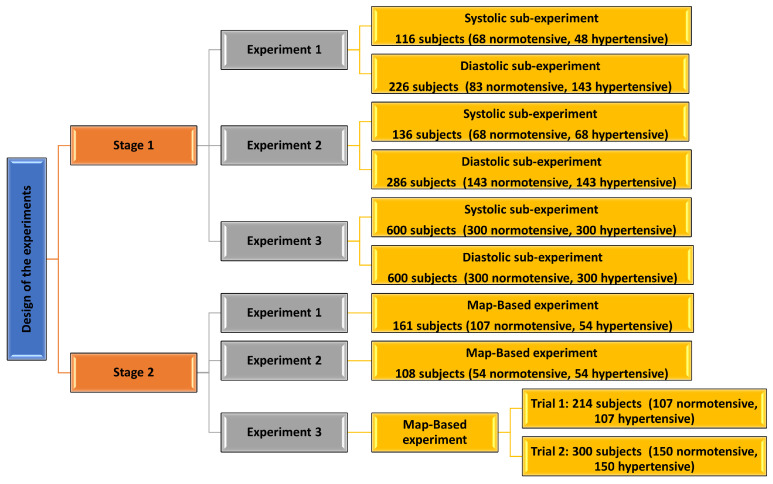
A block diagram showing details of the design of the conducted experiments.

**Table 1 biomedicines-11-00849-t001:** Systolic BP-based dataset demographics statistics.

Class	Subjects Number	Age	Gender	Mean of SBP	Mean of DBP	Mean of BMI
**Normal**	68	(35–61 y), mean = 49.7 ± 6.9	M = 34; F = 34	114.6 ± 5.7 mmHg	78.4 ± 6 mmHg	29.5 ± 6.2
**Hypertensive**	48	(37–62 y), mean = 52.3 ± 5.8	M = 15; F = 33	136.6 ± 7.2 mmHg	85.4 ± 6.4 mmHg	30.6±6.5
**All** **Subjects**	116	(35–62 y), mean = 50.8 ± 6.6	M = 49; F = 67	123.7 ± 12.6 mmHg	81.3 ± 7.1 mmHg	30 ± 6.3

Gender (M = Male; F = Female), SBP: Systolic BP, DBP: Diastolic BP, BMI: Body Mass Index.

**Table 2 biomedicines-11-00849-t002:** Diastolic BP-based dataset demographics statistics.

Class	Subjects Number	Age	Gender	Mean of SBP	Mean of DBP	Mean of BMI
**Normal**	83	(35–61 y), mean = 50.1 ± 7.2	M = 42; F = 41	118.3 ± 8.1 mmHg	74.4 ± 4.3 mmHg	28.5 ± 5.7
**Hypertensive**	143	(35–62 y), mean = 49 ± 6.9	M = 68; F = 75	125.9 ± 9.3 mmHg	84.5 ± 3.9 mmHg	30.2 ± 6
**All** **Subjects**	216	(35–62 y), mean = 49.4 ± 7	M = 110; F = 116	123.1 ± 9.6 mmHg	80.8 ± 6.3 mmHg	29.6 ± 5.9

Gender (M = Male; F = Female), SBP: Systolic BP, DBP: Diastolic BP, BMI: Body Mass Index.

**Table 3 biomedicines-11-00849-t003:** MAP BP-based dataset demographics statistics.

Class	Subjects Number	Age	Gender	Mean of SBP	Mean of DBP	Mean of MAP	Mean of BMI
**Normal**	107	(35–60 y), mean = 49.3 ± 6.3	M = 53; F = 54	124.1 ± 5.3 mmHg	82.2 ± 2.3 mmHg	96.2 ± 1.7 mmHg	29.6 ± 5.6
**Hyper**	54	(36–62 y), mean = 49.8 ± 7.5	M = 25; F = 29	133.8 ± 8.8 mmHg	87.8 ± 4.9 mmHg	103.1 ± 4.4 mmHg	30.9 ± 5.7
**All** **Subjects**	161	(35–62 y), mean = 49.5 ± 6.7	M = 78; F = 83	127.4 ± 8.1 mmHg	84 ± 4.5 mmHg	98.5 ± 4.4 mmHg	30 ± 5.7

Hyper: Hypertensive, Gender (M = Male; F = Female), SBP: Systolic BP, DBP: Diastolic BP, BMI: Body Mass Index.

**Table 4 biomedicines-11-00849-t004:** A sample of feature vectors of four different subjects.

Subject 1	Subject 2	Subject 3	Subject 4
0.444947	0.535083	0.429489	0.467324
0.65227	0.732387	0.636939	0.674185
0.790256	0.846757	0.779007	0.811279
0.874857	0.912191	0.863709	0.893494
0.928818	0.951842	0.922033	0.94358
0.963041	0.978068	0.961477	0.973599
0.984674	0.99284	0.985828	0.990159
0.997025	0.999217	0.99689	0.997966
0.999766	1	0.999308	0.999753
1	0	0.999915	1
0	0	1	0
1.302083	1.171875	1.432292	1.302083
0.329568	0.243342	0.348931	0.301552
12.19939	14.17064	12.98753	13.37305
1.630565	1.571993	1.639687	1.568891
1.068621	1.130165	1.070192	1.061826
0.942445	1.000221	0.929683	0.936494

**Table 5 biomedicines-11-00849-t005:** Stage 1. Systolic BP-based sub-experiment.

Experiment	Systolic BP-Based Sub-Experiment Subjects	Comments on Data
**Exp. 1**	116 (68 normotensive, 48 hypertensive)	Original, unbalanced
**Exp. 2**	136 (68 normotensive, 68 hypertensive)	Original + synthetic, balanced
**Exp. 3**	600 (300 normotensive, 300 hypertensive)	Original + synthetic, balanced

**Table 6 biomedicines-11-00849-t006:** Stage 1. Diastolic BP-based sub-experiment.

Experiment	Diastolic BP-Based Sub-Experiment Subjects	Comments on Data
**Exp. 1**	226 (83 normotensive, 143 hypertensive)	Original, unbalanced
**Exp. 2**	286 (143 normotensive, 143 hypertensive)	Original + synthetic, balanced
**Exp. 3**	600 (300 normotensive, 300 hypertensive)	Original + synthetic, balanced

**Table 7 biomedicines-11-00849-t007:** Stage 2. MAP-based experiments.

Experiment	MAP-Based Experiment Subjects	Data
**Exp. 1**	161 (107 normotensive, 54 hypertensive)	Original, unbalanced
**Exp. 2**	108 (54 normotensive, 54 hypertensive)	Original, balanced
**Exp. 3**	Trial 1: 214 (107 normotensive, 107 hypertensive) Trial 2: 300 (150 normotensive, 150 hypertensive)	Original + synthetic, balanced

**Table 8 biomedicines-11-00849-t008:** The accuracy of classification of different classifiers of the systolic-based sub-experiment (136 subjects).

Classifiers	Kernel	Validation	Accuracy
**Feed-forward**	**hidden-layers = (10, 8)**	**Leave-out**	85%
**Ensemble-Trees**	**RUBoosted**	**15 %**	85%
Ensemble-Trees	Bagged-trees	15 %	80%
SVM	Gaussian	15 %	80%

**Table 9 biomedicines-11-00849-t009:** The accuracy of classification of different classifiers of the diastolic-based sub-experiment (286 subjects).

Classifiers	Kernel/Parameter	Validation	Accuracy
**Decision-Trees**	**Simple**	**20 fold**	72.4%
SVM	Gaussian	20 fold	71.0%
SVM	Gaussian	30 fold	70.6%

**Table 10 biomedicines-11-00849-t010:** The accuracy of classification of different classifiers of the systolic-based sub-experiment (600 subjects).

Classifiers	Kernel/Parameter	Validation-Scenario	Accuracy
**^1^ LR**	**LR**	** 25.0%**	89.3%
SVM	Quadratic	24 fold	84.3%
SVM	Quadratic	48 fold	83.5%

^1^ LR: Logistic Regression.

**Table 11 biomedicines-11-00849-t011:** The accuracy of classification of different classifiers of the diastolic-based sub-experiment (600 subjects).

Classifiers	Kernel/Parameter	Validation	Accuracy
**Feed-forward**	**hidden-layers = (13, 9)**	**Leave-one-out**	88.9%
LR	LR	30 fold	77.3%
^1^ LD	Quadratic	15 fold	76.8%
SVM	Linear	15 fold	76.7%

^1^ LD: Linear Discriminant.

**Table 12 biomedicines-11-00849-t012:** The accuracy of classification of different classifiers of MAP-based experiment (161 subjects (107 normal, 54 hypertensive)).

Classifiers	Kernel/Parameter	Validation-Scenario	Accuracy
**KNN**	**Euclidean**	**10.0 %**	87.5%
KNN	Euclidean	15.0 %	83.3%

**Table 13 biomedicines-11-00849-t013:** The accuracy of classification of different classifiers of MAP-based experiment (108 subjects (balanced (54 normal, 54 hypertensive)).

Classifiers	Kernel/Parameter	Validation-Scenario	Accuracy
**KNN**	**Cosine**	**20.0 %**	71.4%
**SVM**	**Gaussian**	**20.0 %**	71.4%

**Table 14 biomedicines-11-00849-t014:** The accuracy of classification of different classifiers of MAP-based experiment (214 subjects (balanced (107 normal, 107 hypertensive)).

Classifiers	Kernel/Parameter	Validation	Accuracy
**Ensemble**	**AdaBoost**	**10.0 %**	95.2%
KNN	Euclidean/Minkowski	10.0 %	90.5%
SVM	Quadratic/Gaussian	10.0 %	85.7%

**Table 15 biomedicines-11-00849-t015:** The accuracy of classification of different classifiers of MAP-based experiment (300 subjects (balanced (150 normal, 150 hypertensive)).

Classifiers	Kernel/Parameter	Validation	Accuracy
**Ensemble-Trees**	**Bagged-trees**	**10.0 %**	90.0%
**SVM**	**Cubic**	**10.0 %**	90.0%
SVM	Quadratic/Linear	15.0 %	84.4%

**Table 16 biomedicines-11-00849-t016:** SVM classifier results for systolic-based, diastolic-based, and MAP-based experiments (original + synthetic datasets).

Experiment	Number of Subjects	Best Accuracy
**MAP-Based**	**300**	90.0%
Systolic-Based	600	84.3%
Diastolic-Based	600	76.7%

## Data Availability

The data presented in this study are available on request from the corresponding author.

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
