# Peer review of "Using Mean Arterial Pressure in Hypertension Diagnosis versus Using Either Systolic or Diastolic Blood Pressure Measurements"

_biomedicines, 2023, doi:10.3390/biomedicines11030849_

Round 1

Reviewer 1 Report

The comments are presented as follows:

major comments

-          It is not necessary to prepare a long background in the Abstract. please remove the extra sentences.

-          In the abtract please highlight the main novel ideas.

-          please explain clearly what the contribution of authors is and what are the hallmark points of the study at the end of the introduction.

-          please specify the preprocessing section.

-          it is suggested to move the section ‘’Design of experiments’’ before ‘’methods’’

-          please open a subsection at the end of section ‘’method’’ and shortly explain the SVM and KNN concepts with references. Additionally, it is necessary to present one recent significant paper for the SVM and KNN such as

Hekmatmanesh A, Wu H, Jamaloo F, Li M, Handroos H. A combination of CSP-based method with soft margin SVM classifier and generalized RBF kernel for imagery-based brain computer interface applications. Multimedia Tools and Applications. 2020 Jul;79(25):17521-49.

-          Did authors apply statistical analysis to consider the significance of features and accuracy results?

-          it is recommended to explain the reason of higher accuracy mathematical point of view. which parameters or concept in the classifier causes higher accuracy.

minor comments

-          Please don’t use ‘’etc.’’ in the paper, especially in the abstract. it is not a professional word in scientific journals. Here, it is recommended to present the employed classifiers.

-          avoid parentheses as much as possible in the abstract

-          The English language required to be improved. just as an example, ‘’ The 2017 guidelines’’ è ‘’The guidelines in 2017’’. There are more wrong combinations in the paper, please reconsider the complete paper.

-          The section introduction and other sections is 1 paragraph. please divide each of the sections into several paragraphs, it is difficult for the reader to follow the idea.

-          Please introduce the ‘’ SMOTE’’ then use the abbreviation. it is used in 61 but introduced in line 175.

Reviewer 2 Report

This paper compares the classification capability of MAP, Systolic and Diastolic Blood Pressure indices. It claims that it is more beneficial to use MAP than using either systolic or diastolic blood pressure as classification index.

However, this is not a surprising result. MAP is computed based on the information of both indices and is reasonable to carry more information than each index separately. If the authors extend their experiments so that both, and not either, indices are examined, it will not be a surprise if the classification results are even better. The latter is expected since the classifier is given the opportunity to extract a more deep, even an non-liner relationship, between the indices, while MAP is a simple linear combination of the systolic and diastolic pressure. If this is true, I cannot see the contribution of this paper.

Additionally, the experiments reported use different data sets, different classifiers in each table, different parameters, different kernels and even different validation policies. In order to decouple the experimental results and scientific conclusion from the data, the classifiers and the policies, these experiments have to be homogenized. For example, select three to four classifiers, one validation policy and present results for all of them, so that comparison will be more fair.

Reviewer 3 Report

This study uses machine learning methods to justify the links between cerebrovascular changes (extracted from a few MRI features) and blood pressure values (systolic, diastolic and mean value). Although the idea in general is of certain interest, the article is not suitable for publication in its current form. Detailed revision remarks are listed below.  

Major Revision Remarks:

1.       Title: Rephrasing is needed because the title is too long and grammatically irritating: “Comparing the Usefulness of Using ….. Studying”

2.       Abstract: Missing size of the database (type of subjects, number of subjects, number of measurements per subject, measurement conditions, etc.) that is important to justify the proper data for training, validation and test of machine learning classifiers. The numerical results must be linked to either training, validation or test database for proper interpretation. Anyway, independent test is required for reporting the final performance.

3.       Reference list: 24 studies are referenced, only 13 of which were published after 2010! This is a very scarce and insufficient review of the literature, given the rapid development of wearable blood pressure monitoring devices in recent years and the subsequent vast amount of published studies for validation of methods and devices. Papers (Scopus and/or Web of science) presenting either wearable blood pressure sensors as well as artificial intelligence methods for signal and feature processing must be referenced in detail.

4.       Ln 51-55: Aims must highlight the novelty of the study and the expected clinical benefits.

5.       Ln 56-66 “The rest of this paper is organized as follows…”: The overall paragraph should be deleted because it is not necessary to guide the reader how to read an article with standard formatting according to the journal recommendations, including sections: 1. Introduction, 2. Materials and Methods, 3. Results, 4. Discussion, 5. Conclusions. The content of these sections is clear. It is not necessary to introduce methodological details and results as they are incomprehensive at this early stage of the article. Nevertheless, the overlap with previous article should be strictly defined along with the aims of the study.

6.       Section “2. Methods” to become 2. Materials and Methods, including as a first sub-section 2.1. Data Preparation.

7.       2.2. Data preparation: The split of patients into training, validation and test subsets is missing. Do you use data from the same patient for training, validation and test? This issue should be well clarified because it can bias the performance.

8.       Ln 68-80 and Figure 1 must be included as a starting paragraph of the next sub-section 2.2. Methods.

9.       Ln 115: “To categories data based on MAP” -> Clarify the statement.

10.    Ln 117-119: “MAP values were classified into either optimal, normal, high normal, grade 1 hypertension (mild), grade 2 hypertension (moderate), grade 3 hypertension (severe).” -> Six categories are defined, however, the classes in Tables 1 and 2 are only two (Normal and Hypertensive), as well as classifiers are trained with a binary output. Well explain this input annotation inconsistency which strongly affects the output. Include in Tables 1,2,3.  

11.    Tables 1, 2, 3: Add measurement units. Moreover, add measurement units in the text everywhere referring to numerical values (e.g. Ln 119-123, check for others).

12.    Table 4: The feature vectors included in the table are not comprehensively explained in the text. The question is what exactly are these characteristics, what are the measurement units and values range, how many measurements per individual are available?

13.    Figure 4: The white text in white orange and gray font is very faintly visible. Change the colors to a lighter color scheme with black font.

14.    Figure 4: The number of subjects 600 (300 normotensive + 300 hypertensive) is much larger than the patient list in Tables 1,2,3, as well as the text Ln 86: “A data set of 342 subjects of MRA was used in the experimentation.” Explain and correct this discrepancy. Ln 196-218: The experiment was not well described. The synthetic data generation is not well explained too. The synthetic data generation should be described in a separate section.

15.    Figure 4: The overall idea of the experiments and their division in Stage 1 and Stage 2 should be better explained. Unfortunately, the description is a rather incomprehensible mess.

16.    Methods: MISSING Description of Classifiers – their design, input feature space, output type, and hyperparameters!!! RESULTS cannot evaluate and report UNDEFINED methods!

17.    Methods: MISSING definition of performance metrics!!! RESULTS cannot evaluate and report UNDEFINED metrics! They cannot be interpreted in the appropriate context. Be sure to comprehensively explain the methodological background and rationale of the validation method: “Leave-one-out”

18.    Section “3. Experimental Results” -> to become “3. Results”. Note in Ln 220: Simulations in Matlab cannot be referenced to as study experimentation or experimental results. These are just simulation or machine learning numerical results. The term “experimental results” is misleading for real-life experiments on patients and should be corrected all over the text.

19.    Section 3 must be divided to sub-sections according to the study design in Figure 4. Presently, too many results are listed with no comprehensive systematic overview. The explanation is a mixture of bad and not so bad results, and the reader is confused which table to finally consider among the vast amount of Tables: 8, 9, 10, 11, 12, 13, 14, 15, 16. Why so many tables are presented?   

20.    Ln 221-226: Incomprehensive statement. The text must be written in clear English style and well communicated ideas.

21.    Ln 249: “In this manuscript, we have included accuracy values that are ≥ 70.0% only.” -> Clarify the statement. Suggested biased performance reports.

22.    Section Discussion is missing. Presently there is a mix of Results and Discussion, however, the article would be clearer if Discussion is arranged in a separate section, including a clear overview on specific and general results.

23.    Section Discussion: Do you have results from a mixed evaluation of the three values together (Systolic, Diastolic and mean blood pressure) and their gathered impact to Hypertension?

24.    Section Discussion: Comparison with other similar studies is MISSING! The authors must conduct an appropriate literature survey and compare their results to the published ones. This is generally a common report in Discussion.

Round 2

Reviewer 1 Report

The paper can be published with the current format.

Author Response

Dear Reviewer, Thank you so much for your response and support.

Reviewer 2 Report

I would like to thank the authors for the long, analytical reply to my comments.

I have to note, though, that they did not gave a reply to the main point of my comments. I copy below from my initial comments.

"""MAP is computed based on the information of both indices and is reasonable to carry more information than each index separately. If the authors extend their experiments so that both, and

not either, indices are examined, it will not be a surprise if the classification

results are even better.

The latter is expected since the classifier is given the opportunity to extract a

more deep, even an non-liner relationship, between the indices, while MAP is a

simple linear combination of the systolic and diastolic pressure. If this is true, I

cannot see the contribution of this paper."""

In their reply, the authors show that there is an interesting contribution, compared to Kundu et al. However, the review is for the submitted paper and not for Kundu's one. There is a critical point in the submitted paper which limits the scientific value. Even if the authors show that using both SBP and DSP can give better results than using only MAP, this is still not a significant conclusion. The significant conclusion would be to show that machine learning allows better results when only MAP is used, but I cannot see how this can be done and I don't think it is true.

Author Response

Dear reviewer, thank you so much for your reply. The main goal and importance of using MAP from our point of view is to provide a single and simple index to be used in the analysis and investigations of big medical data in a much more easy and more effective way than using 2 indices (SBP, and DBP).

We have mentioned in the introduction  that there exist some studies in the literature that give superior value to SBP measurement over DBP
measurement when diagnosing several hypertension-related diseases based on patients’ age for instance. So, there are claims that using one measurement would be more beneficial than using both.

Besides, the Initial results showed that MAP would be more beneficial and accurate in identifying the cerebrovascular impact of hypertension than just using either systolic BP or diastolic BP. This result emphasizes the pathophysiological significance of MAP and supports prior views that this simple measure may be a superior index for the definition of hypertension and research on hypertension.

Additionally, in your comment, you suggested that using both measurements SBP and DBP would provide more accurate results than using MAP. But the question is what is the method to incorporate both measurements in the analysis. My question is because the only way we know, to the best of our knowledge, to incorporate both measurements in the analysis was by using the MAP equation which uses both measurements in its calculations. If there is another way that we are not aware of, We would  be very willing to use it in our future work. We hope that we could make our point of view clearer and thank you again for your valuable discussion.

Reviewer 3 Report

The authors provided an improved version of their manuscript in response to the reviewer's comments. However, not all comments are correctly or sufficiently addressed. In addition, I provide a list of these comments with relevant follow-up questions and discussion of clarification issues.

Revision Remarks:

1.       Title: Check the title. The article version provided to the reviewer contains the old title and does not correspond to the author’s response: “We suggest to rephrase the title to “Using Mean Arterial Pressure in Hypertension Diagnosis Versus Using Either Systolic or Diastolic Blood Pressure Measurements””.

2.       Both literature research and the reference list are insufficient!!! The authors should provide more and recent references in the related field of blood pressure measurement and its link to clinical cerebrovasculature studies, as well as related machine learning methods for data processing in the focus of this study.

3.       “2.2. Data preparation: The split of patients into training, validation and test subsets is missing. Do you use data from the same patient for training, validation and test? This issue should be well clarified because it can bias the performance.” -> Authors response: “we do not use data from the same patients for training, validation, and test. But rather the partitioning is done between different patients.” -> This issue should be well clarified in the text.

4.       Tables 1, 2, 3: Add measurement units.” -> Authors response: “We have not included the units in the tables due to space constraints and Latex problems” -> You MUST provide correct measurement units of all data in all tables and figures in every scientific paper!!! Latex problems are out of since!!! Authors should find the right way to properly format their tables. Anyway, professional services are available in case the Latex problems persist.

5.       “Table 4: The feature vectors included in the table are not comprehensively explained in the text. The question is what exactly are these characteristics, what are the measurement units and values range, how many measurements per individual are available?” -> Authors response: “This is a sample of the feature vector for 4 subjects and the explanation of these values is mentioned starting from line 173 in the manuscript.” -> I cannot see evidences for your response. Ln 173 starts in section 2.4., while Table 4 is first referred in Section 2.3. Therefore, the reader cannot comprehensively understand the content of Table 4 in the context of the data in section 4.3. Furthermore, section 2.4 (Ln 173) does not refer Table 4 and discusses far different problem than data in Table 4. In my opinion, Table 4 should contain an additional first column including labels for what the successive rows mean. The explanations in the text must refer those labels, as well as the comprehensive link between data in Table 4 and the important image parts in Figure 3.

6.       It is necessary to justify the correctness of the method of generating synthetic data. If this method has been used effectively in a previous study, this should be well explained. Otherwise, more details and rationale behind the method should be provided.

7.       Methods: The new lines Ln 214-238 in section 2.6 Classifiers are TOO GENERAL and look like some guides for lay persons (note about missing references for different classifiers!!!). In fact, the authors NEED to well explain the implementation of their data in the input and settings of hyperparameters of different classifiers. The authors explain in their response to reviewers: “Actually, one classifier can produce different accuracy values with every different scenario of the kernel, parameters, and validation scenario.”, therefore it is IMPORTANT to provide enough numerical values for the classifier settings to produce the same results as in the article. It would be good to include this information in a well-structured table in section 2.6. Classifiers.

8.       Ln 211: The formula should be numbered and formatted according to the journal template requirements.

9.       Caption of Tables 8-16: the term “subjects” has been used. Indeed, this number includes synthetic data but not real subjects (patients). In order not to mislead the reader that data from real patients were used, the authors should replace the terms "subjects" with a more correct term, e.g. "data samples" (or other if more appropriate).

10.    In their answer to reviewers, the authors state that Tables 8-16 are necessary in their extensive study to show the importance of the index MAP. My question is exploring the importance of MAP compared to what? I expected to see а comparative study of MAP vs. both SBP and DBP but in fact section Results does not show such a comparison (or at least it is not comprehensively highlighted). Therefore, it is questionable whether standard SBP and DBP measures would provide better results. Numerical data in answer to this question are important to justify the meaning of this study. These comparative data are also necessary in support of the claim in Ln 299-300: “In this study, we have investigated the importance of the systolic component and diastolic component separately in hypertension diagnosis. The aim was to find out whether one component would be of more value in hypertension detection than the other.” Note that speaking about AIMS in Discussion is out of sense. Discussion MUST directly refer numbered Tables and numerical results.

11.    Authors response: “We have tested the proposed framework using Classification learner in MATLAB R2017a. This version tries so many classifiers with different kernels and parameters. We have also used different feature optimization procedures. We have presented only a subset that produced accuracy >%70. And we did not include other classifiers which produced less accuracy (as they can be easily found through the Classification learner or the MATLAB documentation).” -> The optimization process should be well explained in the text. I currently find no evidence of such an optimization process or any comprehensively described.

Author Response

Dear reviewer, We thank you so much for your extensive review that we benefit from so much. Kindly, find the point to point resonse below:

Revision Remarks:

  1. Title: Check the title. The article version provided to the reviewer contains the old title and does not correspond to the author’s response: “We suggest to rephrase the title to “Using Mean Arterial Pressure in Hypertension Diagnosis Versus Using Either Systolic or Diastolic Blood Pressure Measurements””.

Response: We are sorry for this mistake, we have corrected the title. Thank you so much

  1. Both literature research and the reference list are insufficient!!! The authors should provide more and recent references in the related field of blood pressure measurement and its link to clinical cerebrovasculature studies, as well as related machine learning methods for data processing in the focus of this study.

Response: We have justified this insufficiency because of the lack of studies in this area. The area of hypertension prediction(and NOT diagnosis) using cerebral vascular changes and MAP index Thank you so much

  1. “2.2. Data preparation: The split of patients into training, validation and test subsets is missing. Do you use data from the same patient for training, validation and test? This issue should be well clarified because it can bias the performance.” -> Authors response: “we do not use data from the same patients for training, validation, and test. But rather the partitioning is done between different patients.” -> This issue should be well clarified in the text.

Response: We have added the required clarification at line 242. Thank you

  1. “Tables 1, 2, 3: Add measurement units.” -> Authors response: “We have not included the units in the tables due to space constraints and Latex problems” -> You MUST provide correct measurement units of all data in all tables and figures in every scientific paper!!! Latex problems are out of since!!! Authors should find the right way to properly format their tables. Anyway, professional services are available in case the Latex problems persist.

Response: We have added the measurements units per your valuable recommendation. Thank you so much

  1. “Table 4: The feature vectors included in the table are not comprehensively explained in the text. The question is what exactly are these characteristics, what are the measurement units and values range, how many measurements per individual are available?” -> Authors response: “This is a sample of the feature vector for 4 subjects and the explanation of these values is mentioned starting from line 173 in the manuscript.” -> I cannot see evidences for your response. Ln 173 starts in section 2.4., while Table 4 is first referred in Section 2.3. Therefore, the reader cannot comprehensively understand the content of Table 4 in the context of the data in section 4.3. Furthermore, section 2.4 (Ln 173) does not refer Table 4 and discusses far different problem than data in Table 4. In my opinion, Table 4 should contain an additional first column including labels for what the successive rows mean. The explanations in the text must refer those labels, as well as the comprehensive link between data in Table 4 and the important image parts in Figure 3.

Response: We are sorry for this misunderstanding. The lines that explain the values in table start at line 162 instead of 173. As explained these feature vectors are the results of computations of the two procedures that compute the change in diameter and change in tortuosity. The change in diameters is represented by probability density functions (PDF) and Cumulative distribution functions. And the first 11 value of each vector represents the PDF of each diameter of blood vessels and the rest of the values represent the change of tortuosity in blood vessels in terms of the medians and means of both MEAN curvature and Gaussian curvature), we have taken just the medians and averages of the curvature to simplify the analysis instead of using the h=very huge files that were computed at every single point along each blood vessel inside the brain. Thank you so much.

  1. It is necessary to justify the correctness of the method of generating synthetic data. If this method has been used effectively in a previous study, this should be well explained. Otherwise, more details and rationale behind the method should be provided.

Response: SMOTE is a very common, simple, and effective algorithm that generates synthetic data samples that takes the features of the original data. We have added references to other studies that used SMOTE. Please refer to line 171. Thank you.

  1. Methods: The new lines Ln 214-238 in section 2.6 Classifiers are TOO GENERAL and look like some guides for lay persons (note about missing references for different classifiers!!!). In fact, the authors NEED to well explain the implementation of their data in the input and settings of hyperparameters of different classifiers. The authors explain in their response to reviewers: “Actually, one classifier can produce different accuracy values with every different scenario of the kernel, parameters, and validation scenario.”, therefore it is IMPORTANT to provide enough numerical values for the classifier settings to produce the same results as in the article. It would be good to include this information in a well-structured table in section 2.6. Classifiers.

Response: Thank you so much for your comment. These classifiers are very common and standard in the engineering field and usually, they are not explained extensively in similar studies. Besides, it is very challenging to mention all the scenarios we have tried with all parameters. This would require a book and not just an article paper and will not be useful to the user or the goal of the paper. We have also mentioned that we have used the classification learner in MATLAB, so that any reader can find the standard associated parameters of each classifier easily.

  1. Ln 211: The formula should be numbered and formatted according to the journal template requirements.

Response. Thank you for your comment, we have fixed this issue.

  1. Caption of Tables 8-16: the term “subjects” has been used. Indeed, this number includes synthetic data but not real subjects (patients). In order not to mislead the reader that data from real patients were used, the authors should replace the terms "subjects" with a more correct term, e.g. "data samples" (or other if more appropriate).

Response. Thank you for your comment, we believe that “data samples” would not be accurate too because the data includes the original data sets. We have fixed that by determining how many data are synthetic and how many are original starting at line 196 in the manuscript.

  1. In their answer to reviewers, the authors state that Tables 8-16 are necessary in their extensive study to show the importance of the index MAP. My question is exploring the importance of MAP compared to what? I expected to see Ð° comparative study of MAP vs. both SBP and DBP but in fact section Results does not show such a comparison (or at least it is not comprehensively highlighted). Therefore, it is questionable whether standard SBP and DBP measures would provide better results. Numerical data in answer to this question are important to justify the meaning of this study. These comparative data are also necessary in support of the claim in Ln 299-300: “In this study, we have investigated the importance of the systolic component and diastolic component separately in hypertension diagnosis. The aim was to find out whether one component would be of more value in hypertension detection than the other.” Note that speaking about AIMS in Discussion is out of sense. Discussion MUST directly refer numbered Tables and numerical results.

Response: thank you so much for your comment. The main goal and importance of using MAP from our point of view is to provide a single and simple index to be used in the analysis and investigations of big medical data in a much more easy and more effective way than using 2 indices (SBP, and DBP).

We have mentioned in the introduction  that there exist some studies in the literature that give superior value to SBP measurement over DBP measurement when diagnosing several hypertension-related diseases based on patients’ age for instance. So, there are claims that using one measurement would be more beneficial than using both.

Besides, the Initial results showed that MAP would be more beneficial and accurate in identifying the cerebrovascular impact of hypertension than just using either systolic BP or diastolic BP. This result emphasizes the pathophysiological significance of MAP and supports prior views that this simple measure may be a superior index for the definition of hypertension and research on hypertension.

Additionally, in your comment, you suggested that using both measurements SBP and DBP would provide more accurate results than using MAP. But the question is what is the method to incorporate both measurements in the analysis? My question is because the only way we know, to the best of our knowledge, to incorporate both measurements in the analysis was by using the MAP equation which uses both measurements in its calculations. If there is another way that we are not aware of, We would  be very willing to use it in our future work. We hope that we could make our point of view clearer and thank you again for your valuable discussion.

  1. Authors response: “We have tested the proposed framework using Classification learner in MATLAB R2017a. This version tries so many classifiers with different kernels and parameters. We have also used different feature optimization procedures. We have presented only a subset that produced accuracy >%70. And we did not include other classifiers which produced less accuracy (as they can be easily found through the Classification learner or the MATLAB documentation).” -> The optimization process should be well explained in the text. I currently find no evidence of such an optimization process or any comprehensively described.

Response: We have not used optimization techniques in the current study. We have relied on the numerical accuracy values of each experiment. But we will make the optimization of our experiments in our future plans. Thank you so much.

Round 3

Reviewer 2 Report

The authors may use my recommendation as a future work as they suggested. This could be an interesting extension for this work, I believe. The linear relation between the systolic and diastolic blood pressure seems a weak correlation, while machine learning can be used for identifying a deeper relation.